# Effect of PTPN22, FAS/FASL, IL2RA and CTLA4 genetic polymorphisms on the risk of developing alopecia areata: A systematic review of the literature and meta-analysis

S. R. Gil-Quiñones[1ᴑ], I. T. Sepúlveda-Pachón[1], G. Sánchez Vanegas[2], L. D. Gutierrez-Castañeda[3ᴑ]*

**1** Clinical Epidemiology Program, Fundación Universitaria de Ciencias de la Salud (FUCS), Bogotá, Colombia, **2** Clinical Epidemiology Program, Research Institute, Fundación Universitaria de Ciencias de la Salud (FUCS), Bogotá, Colombia, **3** Research Institute, Group of Basic Sciences in Health (CBS)-FUCS, Fundación Universitaria de Ciencias de la Salud (FUCS), Bogotá, Colombia

ᴑ These authors contributed equally to this work.
* ldgutierrez@fucsalud.edu.co

## Abstract

### Objectives

Genetic association studies on alopecia areata (AA) performed in various populations have shown heterogeneous results. The aim of the current review was to synthesize the results of said studies to estimate the impact of *FAS*, *FASL*, *PTPN22*, *CTLA4* and *IL2RA* gene polymorphisms on AA susceptibility.

### Design

A systematic literature search was conducted in the Medline, Web of Science, Scopus, EMBASE and LILACS databases. Studies published up to June 2020 were included. The results available in the grey literature including the Open Grey and Google Scholar databases were also used. The texts of potentially related studies were screened by individual reviewers. Evidence of publication bias was assessed using the Newcastle-Ottawa scale and the quality of evidence was assessed using the GRADE system. The quantitative synthesis was performed using the fixed effect model.

### Results

Out of 1784 articles, we identified 18 relevant articles for the qualitative synthesis and 16 for the quantitative synthesis. In a study of rs2476601 polymorphism of *PTPN22* gene, including 1292 cases and 1832 controls, a correlation was found with the risk of developing AA in the allelic model (OR1.49 [95% C:1.13–1.95]), the heterozygous codominant (OR1.44 [95% CI:1.19–1.76]) and dominant model (OR1.43 [95% CI:1.18–1.73]). No association was found between the presence of *FASL*, *PTPN22*, *CTLA* and *IL2RA* gene polymorphisms with AA susceptibility.

**Data Availability Statement:** All relevant data are within the manuscript and its Supporting information files.

**Funding:** The authors thank FUNDACION UNIVERSITARIA DE CIENCIAS DE LA SALUD for financial support.

**Competing interests:** The authors have declared that no competing interests exist.

## Conclusions

The results suggest that the *T* allele of the single nucleoid polymorphism (SNP) rs2476601 in *PTPN22* gene is a risk factor for developing alopecia areata. However, more robust studies defining the ethnic background of the population of origin are required, so that the risk identified in the present study can be validated. Additionally, a greater number of studies is necessary to evaluate the role of the *FAS*, *FASL*, *PTPN22*, *CTLA4* and *IL2RA* genetic variants, given the heterogenous results found in the literature.

## Introduction

Alopecia areata (AA) is a multifactorial disease in which environmental, neuro-endocrinological, immunological and genetic factors are involved [1]. AA affecting the hair follicle is the most common form, through a breakdown in immune privilege of the hair follicle in the anagen (hair growth) phase by CD4$^+$ and CD8$^+$ T lymphocytes, resulting in non-scarring hair loss [2]. AA encompasses a spectrum of disease patterns including patchy alopecia (diffuse loss) alopecia totalis (total hair loss on the scalp) and alopecia universalis (loss of hair on the entire body) [3].

Given the broad genetic component of AA, the effect of several immune response modulator genes has been discussed [4]. Genome-wide association studies (GWAS) have revealed the involvement of genes related to innate and adaptive immunity. These population studies have proposed that the *PTPN22*, *FAS*, *FASL*, *IL2RA* and *CTLA4* genes are related to the risk of developing AA [5].

The *FAS* gene encodes tumor necrosis factor receptor superfamily member 6 and *FASL* gene encodes tumor necrosis factor ligand superfamily member 6, located on chromosomes 10 and 1 respectively. Their function is critical in immunological homeostasis given their ability to induce cell death and proliferation or differentiation of T lymphocytes [6]. The *PTPN22* gene is located on chromosome 1 and encodes a protein involved in T lymphocyte signaling, downregulating the T cell receptor and the production of type 1 interferon [7]. *CTLA4* gene encodes a lymphocyte receptor that promotes T-cell anergy, preventing autoimmune reactions [8]. Finally, *IL2RA* is located on chromosome 10 and encodes one of the subsets of the IL-2 receptor, which is involved in the regulation of immunological tolerance and the control of regulatory T cells [9].

Several studies have reported an association between the aforementioned genes and the risk of developing AA. A case-control study carried out by Kalkan et al., which investigated *FAS*-670*A/G* (rs1800682) and *FASL*-124*A/G* (rs5030772) polymorphisms in the Turkish population, found that the GG genotype of rs1800682 polymorphism was a protective factor against AA with a reduced risk of AA compared with the *AA* and *AG* genotypes (OR 0.07 [95% CI: 0.00–0.41]) [10]. The *PTPN22* gene and the *CT* genotype of rs2476601 polymorphism, were associated with AA susceptibility (OR 3.31 [95% CI: 1.008–10.892]) [11]. A haplotype analysis of *CTLA-4* determined that the presence of alleles A (rs231775) and G (*CT*60: rs3087243) is associated with a lower risk of the disease (OR 0.28 [95% CI: 0.09–0.82]) [12]. An association analysis of the *IL2RA* gene in the Chinese population, determined that the prevalence of rs3118470 polymorphism genotype in the AA group was 48.2% for *T/C*, 35.6% for *T/T*, and 16.2% for *C/C*. No Odds Ratio risk estimator was calculated [13].

The results of primary studies of genetic association between polymorphisms in said genes with AA susceptibility in different populations, have yielded heterogeneous and sometimes contradictory results. For these reasons, the present study synthesizes the available evidence regarding *PTPN22*, *FAS*, *FASL*, *IL2RA* and *CTLA4* gene polymorphisms and their correlation with AA susceptibility.

## Methods

### Inclusion and exclusion criteria

This review was based on the PRISMA guidelines (S1 File). Studies should satisfy the following criteria: a) Human population studies; b) Genetic association studies on AA risk and *FAS/FASL*, *PTPN22*, *CTLA4* and *IL2RA* gene polymorphisms; c) Studies in patients diagnosed with any type of AA (patches/totalis/universalis). The studies should be genome-wide associated studies (case-control/GWAS case-control studies). The search was conducted without language or year of publication restrictions.

The exclusion criteria were: control patients presenting with an autoimmune diseases (since their presence may alter the interpretation of the effect of polymorphism on AA susceptibility) and studies excluded for ethical concerns or distortion of scientific results.

### Search strategy

An exhaustive search conducted in the Medline, Web of Science, Scopus, EMBASE and LILACS databases by combining search strategies using the PICO elements of the research question (for observational studies): "Gene (*FAS-FASL/PTPN22/CTLA4/IL2RA*)", "Polymorphism / Genetic Variant" and "Alopecia areata". Studies published up to June 2020 were included. A literature search was conducted for each gene and its homonyms. Other potentially useful resources were identified in Open Grey and Google Scholar. The complete search strategy is fully stated in S2 File. The records were downloaded to a reference software and duplicates were eliminated. Two authors independently selected eligible studies (SG/IS) by reading the title and abstract. Subsequently, the full text of potentially relevant articles was reviewed and meticulously examined for compliance with the inclusion and exclusion criteria using a pre-established form. Discrepancies found in these phases of the process, were resolved by discussion with consensus and checked by a third author (GS). The reasons for exclusion of studies were documented.

### Data extraction

Data was extracted by two authors (SG/IS) using a preestablished form, which included highly relevant information to develop the synthesis, such as: article reference, year of publication, number of cases and controls, Hardy-Weinberg equilibrium, clinical diagnosis, sequencing method, gene analyzed, allelic and genotypic frequency, clinical significance of the variant analyzed, source of funding and conflicts of interest. Authors were contacted via email to request missing information. Discrepancies in extracted data were resolved by a third author (LG).

### Assessment of risk of bias and quality of evidence for individual studies

The Newcasttle-Ottawa scale (NOS) scoring system (0–9) was used to determine the risk of bias for each individual study. Domains inherent to case-control studies, such as case/control selection, comparability, and exposure measurement were evaluated. In accordance with the trend in the scientific community for this type of study, a cut-off ≥7 was established to consider low risk of bias. The methodological quality of the primary studies was assessed using the

NOS and the quality of the evidence found was established using the GRADE tool (). This evaluation was carried out by 2 authors (SG/IS) and discrepancies were resolved by a third author (GS).

## Statistical analysis

All statistical analyses were performed using Stata16$^{\circledR}$ and Revman V5.2 software. Testing for Hardy-Weinberg equilibrium was verified in the control group of each study, if not detected, this calculation was made using a chi-squared test ($X^2$). Odds Ratio (OR) was calculated for dichotomous data using the 2x2 table. The summary measure of the effect was calculated as an Odds Ratio with its respective 95% confidence interval and p-value (a p value <0.05 is considered statistically significant).

The heterogeneity across the included studies was visually assessed based on the results of the forest plots, the chi-squared test for heterogeneity ($X^2$) with a statistical significance value of 10%, the $H^2$ test and the I-squared ($I^2$) statistic. A fixed effect model assuming an $I^2$ statistic of less than 50% was used for the quantitative synthesis. The random effect model was calculated, and subgroup analyses were performed when the $I^2$ score was above 50%, considered to indicate substantial clinical heterogeneity. Heterogeneity was addressed, by performing subgroup analyses across studies according to the NOS score and control group selection (hospital or community-based controls). Meta-regression was applied to determine possible heterogeneity derived from the number of subjects included in the studies. When the heterogeneity value was greater than 75%, it was decided to exclude these studies from the meta-analysis.

## Results

### Results of search

The flow chart of literature search and screening is shown in Fig 1. By means of the combination of terms listed in the search strategy, carried out in the different databases, a total of 1774 articles were obtained, to which 10 articles regarding the application of the snowball strategy to relevant articles, were added. We identified and removed 1338 duplicate records from the 1784 records, obtaining 446 eligible articles for screening by title and abstract. According to the application of the selection criteria 426 irrelevant records were removed. Finally, 20 articles were included for assessment of their full text. Eighteen articles were included in the qualitative synthesis and 16 in the meta-analysis.

### Study characteristics

A total of 17 case-control studies and one case-control GWAS were included. Table 1 contains the characteristics of the included studies. The distribution of the studies according to the gene and genetic variants was: 7/18 for the PTPN22 gene (rs2476601), 4/18 for the *FAS* gene (rs1800682), 3/18 for the *FASL* gene (rs5030772), 3/18 for the *IL2RA* gene (rs3118470) and 4/18 for the *CTLA4* gene (rs231775). These studies were published between 2008 and 2020 and were conducted mainly in geographic regions of Europe and Asia, followed by North America and Mexico to a lesser extent. Control groups composition was community or control-based. The main genotyping method was PCR-RFLP (Polymerase chain reaction—Restriction fragment length polymorphisms). The NOS score ranged between 8 and 9. The specific scores for each domain of bias are listed in Table 2. The studies included the 3 types of alopecia: patchy, totalis and universalis. Allelic and genotypic frequencies are shown in Table 3.

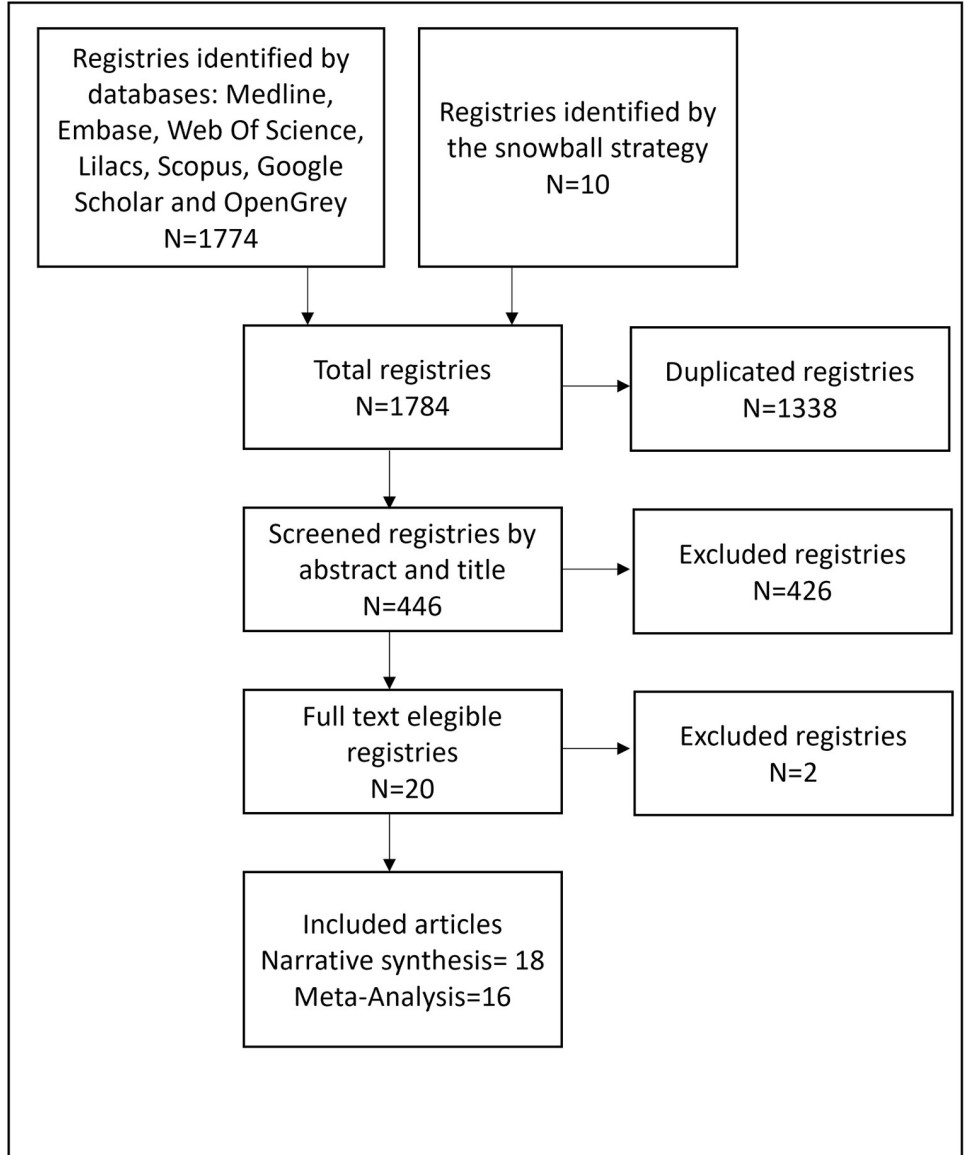

**Fig 1. Study flow diagram.**

## Assessment of risk of bias and quality of evidence

A study with an NOS score between 8 and 9 has high quality (Table 2). The GRADE score approach was used to grade the quality of associations such as *PTPN22* with AA. A moderate quality of evidence was observed for the *T* vs *C* allelic model and a high quality of evidence for the heterozygous and dominant co-dominant models (S3 File). The graphic and statistical evaluation of publication bias was not performed since the studies included were less than 10.

## Association of PTPN22 with the risk of developing alopecia areata

The combined analysis of 1292 cases and 1932 controls showed a statistically significant association between SNP rs2476601 polymorphism of *PTPN22* gene and the risk of developing AA.

**Table 1. Characteristics of the studies included in the systematic review and meta-analysis.**

| First Author | Year | Gene | Variant | Geographic Region | Study design | Control origin | NOS* | #Cases | #Controls |
|---|---|---|---|---|---|---|---|---|---|
| Betz RC [14] | 2007 | PTPN22 | rs2476601 | Germany/Belgium | Case-control | Community | 8 | 435 | 628 |
| Bhanusali D [15] | 2014 | PTPN22 | rs2476601 | United States | Case-control | Community | 8 | 365 | 273 |
| El-Zawahry B [16] | 2013 | PTPN22 | rs2476601 | Egypt | Case-control | Community | 9 | 103 | 100 |
| Kemp E [17] | 2006 | PTPN22 | rs2476601 | England | Case-control | Unclear | 8 | 196 | 507 |
| Moravvej H [4] | 2018 | PTPN22 | rs2476601 | Iran | Case-control | Community | 9 | 69 | 69 |
| Salinas-Santander M [11] | 2015 | PTPN22 | rs2476601 | Mexico | Case-control | Community | 9 | 64 | 225 |
| Shehata W [18] | 2020 | PTPN22 | rs2476601 | Egypt | Case-control | Community | 9 | 60 | 30 |
| Fan X [19] | 2010 | FAS | rs1800682 | China | Case-control | Hospital | 8 | 84 | 84 |
| Kalkan G [10] | 2013 | FAS | rs1800682 | Turkey | Case-control | Hospital | 8 | 118 | 118 |
| Seleit I [20] | 2018 | FAS | rs1800682 | Egypt | Case-control | Unclear | 8 | 60 | 40 |
| Tabatabaei-Panah P [21] | 2020 | FAS | rs1800682 | Iran | Case-control | Hospital | 8 | 60 | 60 |
| Kalkan G [10] | 2013 | FASL | rs5030772 | Turkey | Case-control | Hospital | 8 | 118 | 118 |
| Seleit I [20] | 2018 | FASL | rs5030772 | Egypt | Case-control | Unclear | 8 | 60 | 40 |
| Tabatabaei-Panah P [21] | 2020 | FASL | rs5030772 | Iran | Case-control | Hospital | 8 | 60 | 60 |
| Miao Y [13] | 2013 | IL2RA | rs3118470 | China | Case-control | Hospital | 8 | 427 | 430 |
| Moravvej H [4] | 2018 | IL2RA | rs3118470 | Irán | Case-control | Community | 9 | 69 | 69 |
| Redler S [22] | 2012 | IL2RA | rs3118470 | Germany/Belgium | Case-control | Community | 9 | 768 | 658 |
| Ismail N [23] | 2020 | CTLA4 | rs231775 | Egypt | Case-control | Hospital | 8 | 93 | 93 |
| John K [24] | 2011 | CTLA4 | rs231775 | Central Europe | GWAS Case-control | Unclear | 8 | 1196 | 1280 |
| Megiorni F [12] | 2013 | CTLA4 | rs231775 | Italy | Case-control | Hospital | 8 | 130 | 189 |
| Salinas-Santander M [25] | 2020 | CTLA4 | rs231775 | Mexico | Case-control | Hospital | 8 | 50 | 100 |

Variants described as RS code.

*Newcastle-Ottawa Scale (NOS) Score.

The presence of the *T* allele (rs2476601) was associated with the development of AA (OR1.49 [95% CI:1.13–1.95] $p$ = 0.004) when using the allelic model (Fig 2). A subgroup analysis considering the NOS score was conducted to study the heterogeneity found in the model. A greater association was found between rs2476601 (*PTPN22*) and AA in the subgroup with a score of 9 (OR 2.22 [95% CI:1.16–4.24]) compared to the group with a score of 8 (OR 1.31 [95% CI:1.08–1.58]). The greatest heterogeneity was found in the group with a score of 9 ($I^2$: 55.72%). A significant association was found (OR1.66 [95% CI:1.11–2.48]) in the community-based control groups (S4 and S5 Files) in the control group selection subgroup analysis.

A combined analysis was performed for the homozygous codominant (*TT* vs *CC*), heterozygous codominant (*CT* vs *CC*), dominant (*TC + TT* vs *CC*) and recessive (*TT* vs *CC + CT*) models. The analysis for rs2476601 polymorphism of *PTPN22* gene evidenced that the risk of AA was 1.44-fold greater in subjects with the *CT* genotype compared with the *CC* genotype (OR 1.44 [95% CI: 1.18–1.76] $p$ = 0.000) Fig 3. The dominant model (*TC + TT* vs *CC*) yielded an (OR of 1.43 [95% CI:1.18–1.73] $p$ = 0.000) (Fig 4). The associations obtained for the homozygous codominant models and for the recessive model were not significant (Table 4).

## Association of *FAS* and *FASL* gene polymorphisms with the risk of developing alopecia areata

A meta-analysis of the allelic model for SNP rs1800682 FAS polymorphism, showed that the combined measure of the effect obtained from the allelic model (*G* vs *A*) was not significant (OR 0.9 [95% CI, 0.60–1.34] $p$ = 0.598) (S5 File). To explore heterogeneity, a subgroup analysis

**Table 2. Risk of evaluation bias of included studies using the Newcastle-Ottawa scale.**

| Study | | Newcastle-Ottawa Domains | | | | | | | | |
|---|---|---|---|---|---|---|---|---|---|---|
| | | Selection | | | | Comparability | Exposure | | | Total Score |
| First Author | Year | Adequate case definition | Representativeness of the cases | Selection of control | Definition of control | Control of important confusion factors | Ascertainment of exposure | Same method of ascertainment for cases and controls | Non-Response rate | |
| Betz RC | 2008 | 1 | 1 | 1 | 0 | 2 | 1 | 1 | 1 | 8 |
| Bhanusali D | 2013 | 1 | 1 | 0 | 1 | 2 | 1 | 1 | 1 | 8 |
| El-Zawahry B | 2013 | 1 | 1 | 1 | 1 | 2 | 1 | 1 | 1 | 9 |
| Kemp E | 2006 | 1 | 1 | 0 | 1 | 2 | 1 | 1 | 1 | 8 |
| Moravvej H | 2018 | 1 | 1 | 1 | 1 | 2 | 1 | 1 | 1 | 9 |
| Salinas-Santander M | 2015 | 1 | 1 | 1 | 1 | 2 | 1 | 1 | 1 | 9 |
| Shehata W | 2020 | 1 | 1 | 1 | 1 | 2 | 1 | 1 | 1 | 9 |
| Fan X | 2010 | 1 | 1 | 0 | 1 | 2 | 1 | 1 | 1 | 8 |
| Kalkan G | 2013 | 1 | 1 | 0 | 1 | 2 | 1 | 1 | 1 | 8 |
| Seleit I | 2018 | 1 | 1 | 0 | 1 | 2 | 1 | 1 | 1 | 8 |
| Tabatabaei-Panah P | 2020 | 1 | 1 | 0 | 1 | 2 | 1 | 1 | 1 | 8 |
| Miao Y | 2014 | 1 | 1 | 0 | 1 | 2 | 1 | 1 | 1 | 8 |
| Moravvej H | 2018 | 1 | 1 | 1 | 1 | 2 | 1 | 1 | 1 | 9 |
| Redler S | 2012 | 1 | 1 | 1 | 1 | 2 | 1 | 1 | 1 | 9 |
| Ismail N | 2020 | 1 | 1 | 0 | 1 | 2 | 1 | 1 | 1 | 8 |
| John K | 2011 | 1 | 1 | 0 | 1 | 2 | 1 | 1 | 1 | 8 |
| Megiorni F | 2013 | 1 | 1 | 0 | 1 | 2 | 1 | 1 | 1 | 8 |
| Salinas-Santander M | 2020 | 1 | 1 | 0 | 1 | 2 | 1 | 1 | 1 | 8 |

was performed based on control group selection criteria. Subgroup analysis of the NOS score was not performed since all had a score of 8.

The association obtained for the dominant model was not significant (OR1.03 [95% CI: 0.55–1.96] $p$ = 0.917). We decided not to include the homozygous codominant, heterozygous codominant and recessive models in our meta-analysis for a substantial heterogeneity was detected (Table 4).

The meta-analysis of the allelic model for SNP rs5030772 FASL polymorphism showed a not statistically significant (OR 1.57 [95% CI: 0.91–2.71] $p$ = 0.108) (S5 and S6 Files) combined measure of the effect for the allelic approach ($G$ vs $A$). When performing the subgroup analysis based on control group selection criteria, it was evidenced that the heterogeneity found was mainly given by the group of hospital-based controls. As for the results obtained for SNP rs1800682 FAS polymorphism, all the studies on rs5030772 FASL polymorphism had a NOS score of 8, thus, it was not possible to explain this parameter using a subgroup study analysis.

Regarding the combined analysis for the different genetic models, the association obtained for the recessive model was not statistically significant (OR 0.52 [95% CI: 0.16–1.70] $p$ = 0.282). It was decided not to include the homozygous codominant, heterozygous codominant and dominant models in the meta-analysis, for a substantial heterogeneity was observed (Table 4).

**Table 3. Case-control allelic and genotypic frequencies.**

| Study information | | | | | Type of alopecia areata | | | | Cases | | | | | Controls | | | | |
|---|---|---|---|---|---|---|---|---|---|---|---|---|---|---|---|---|---|---|
| Author | Year | Gene (RS) | Major Allele | Minor Allele | Patchy | Totalis | Universalis | AT/AU | Cases | Risk Alelle | WT | HT | HH | Controls | Risk Alelle | WT | HT | HH |
| Betz RC | 2008 | PTPN22 (rs2476601) | C | T | 196 | . | . | 239 | 435 | 126 | 320 | 104 | 11 | 628 | 132 | 506 | 112 | 10 |
| Bhanusali D | 2013 | PTPN22 (rs2476601) | C | T | 194 | 78 | 93 | 171 | 365 | 69 | 296 | 69 | 0 | 273 | 48 | 225 | 48 | 0 |
| El-Zawahry B | 2013 | PTPN22 (rs2476601) | C | T | 103 | 0 | 0 | 0 | 103 | 23 | 84 | 15 | 4 | 100 | 8 | 92 | 8 | 0 |
| Kemp E | 2006 | PTPN22 (rs2476601) | C | T | 107 | . | . | 84 | 196 | 41 | 155 | 41 | 0 | 507 | 86 | 425 | 79 | 3 |
| Moravvej H | 2018 | PTPN22 (rs2476601) | C | T | 69 | 0 | 0 | 0 | 69 | 24 | 50 | 14 | 5 | 69 | 23 | 55 | 5 | 9 |
| Salinas-Santander M | 2015 | PTPN22 (rs2476601) | C | T | 62 | 1 | 1 | 2 | 64 | 5 | 59 | 5 | 0 | 225 | 7 | 218 | 7 | 0 |
| Shehata W | 2020 | PTPN22 (rs2476601) | C | T | . | . | . | . | 60 | 36 | 32 | . | . | 30 | 6 | 25 | . | . |
| Fan X | 2010 | FAS (rs1800682) | A | G | . | . | . | . | 84 | 61 | 36 | 35 | 13 | 84 | 75 | 22 | 49 | 13 |
| Kalkan G | 2013 | FAS (rs1800682) | A | G | 118 | 0 | 0 | 0 | 118 | 81 | 37 | 81 | 0 | 118 | 91 | 40 | 65 | 13 |
| Seleit I | 2018 | FAS (rs1800682) | A | G | 30 | 15 | 15 | 30 | 60 | 65 | 9 | 37 | 14 | 40 | 31 | 13 | 23 | 4 |
| Tabatabaei-Panah P | 2020 | FAS (rs1800682) | A | G | . | . | . | . | 60 | 68 | 16 | 20 | 24 | 60 | 80 | 16 | 8 | 36 |
| Kalkan G | 2013 | FASL (rs5030772) | A | G | 118 | 0 | 0 | 0 | 118 | 42 | 78 | 38 | 2 | 118 | 43 | 40 | 65 | 13 |
| Seleit I | 2018 | FASL (rs5030772) | A | G | 30 | 15 | 15 | 30 | 60 | 50 | 18 | 34 | 8 | 40 | 22 | 24 | 10 | 6 |
| Tabatabaei-Panah P | 2020 | FASL (rs5030772) | A | G | . | . | . | . | 60 | 32 | 32 | 24 | 4 | 60 | 16 | 48 | 8 | 4 |
| Miao Y | 2014 | IL2RA (rs3118470) | T | C | . | . | . | . | 427 | 510 | 69 | 206 | 152 | 430 | 592 | 27 | 214 | 189 |
| Moravvej H | 2018 | IL2RA (rs3118470) | T | C | 69 | 0 | 0 | 0 | 69 | 44 | 39 | 16 | 14 | 69 | 16 | 59 | 5 | 5 |
| Redler S | 2012 | IL2RA (rs3118470) | T | C | 303 | 465 | 0 | 0 | 768 | 952 | . | . | . | 658 | 895 | . | . | . |
| Ismail N | 2020 | CTLA4 (rs231775) | A | G | 93 | 0 | 0 | 0 | 93 | 73 | 34 | 45 | 14 | 93 | 82 | 19 | 66 | 8 |
| John K | 2011 | CTLA4 (rs231775) | A | G | . | . | . | . | 1196 | 496 | . | . | . | 1280 | 462 | . | . | . |
| Megiorni F | 2013 | CTLA4 (rs231775) | A | G | 71 | 30 | 29 | 59 | 130 | 92 | 52 | 64 | 14 | 189 | 139 | 75 | 89 | 25 |
| Salinas-Santander M | 2020 | CTLA4 (rs231775) | A | G | 45 | 1 | 4 | 5 | 50 | 46 | 15 | 24 | 11 | 100 | 93 | 28 | 50 | 21 |

AT/AU: Alopecia totalis + Alopecia Universalis; WT: Wild Type genotype; HT: Heterozygous genotype; HH: Homozygous genotype.

## Association analysis of the CTLA4 gene with the risk of developing alopecia areata

When evaluating the relation between *CTLA4* rs231775 polymorphism and the risk of developing AA, the association obtained from the allelic model was not statistically significant (OR 1.1 [95% CI: 0.98–1.24] *p* = 0.108) (S6 and S7 Files).

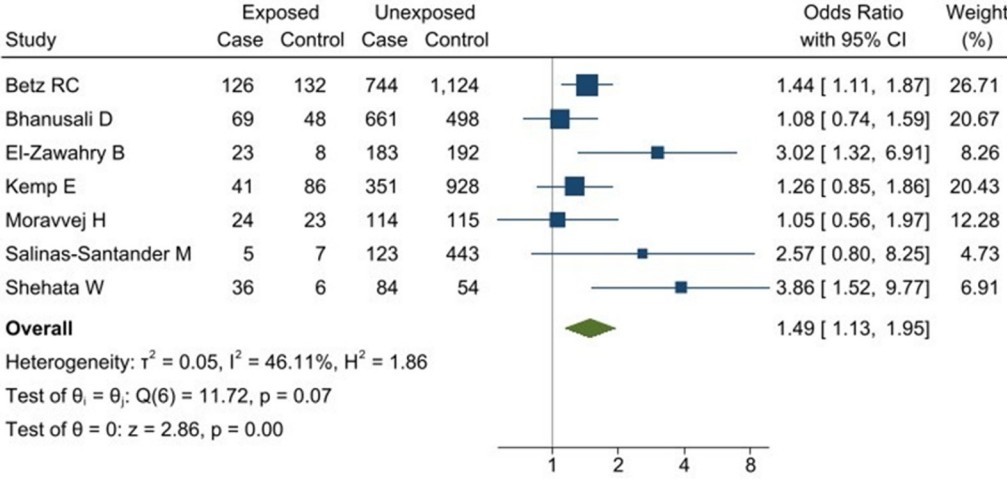

**Fig 2. Association between rs2476601 polymorphism of *PTPN22* gene and the risk of alopecia areata.** Allelic model T vs C. **Exposed group:** patients with allele T; **Unexposed group:** patients with allele *C*.

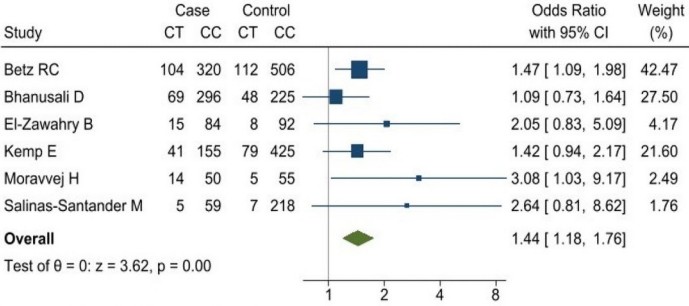

**Fig 3. Association between rs2476601 polymorphism of *PTPN22* gene and risk of alopecia areata.** Codominant model (*CT* vs *CC*).

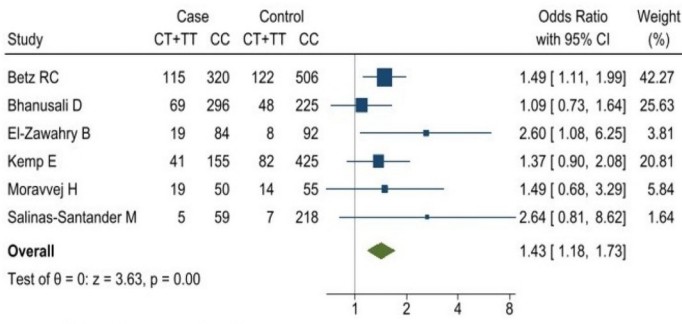

**Fig 4. Association between rs2476601 polymorphism of *PTPN22* gene and risk of alopecia areata.** Dominant model (*CT+TT* vs *CC*).

**Table 4. Association between genes *PTPN22*, *FAS/FASL* and *CTLA4* with alopecia areata.**

| Gen | Model | Allele/Genotype | Association | | | Meta-Analysis model | Heterogeneity | |
|---|---|---|---|---|---|---|---|---|
| | | | OR | IC-95% | p-value | Effect | I2% | *p*-value het |
| *PTPN22* (rs2476601) | **Allelic** | T/C | 1.49 | 1.13–1.95 | **0.004*** | Random | 46.11 | 0.068 |
| | **Codominant** | CC | 1 | - | - | - | - | - |
| | | CT | 1.44 | 1.18–1.76 | **0.000*** | Fixed | 0 | 0.387 |
| | | TT | 1.33 | 0.73–2.43 | 0.349 | Fixed | 13.25 | 0.434 |
| | **Dominant** | CT + TT vs CC | 1.43 | 1.18–1.73 | **0.000*** | Fixed | 0 | 0.464 |
| | **Recessive** | TT vs CC +CT | 1.2 | 0.66–2.18 | 0.549 | Fixed | 18.33 | 0.389 |
| *FAS* (rs1800682) | **Allelic** | G/A | 0.9 | 0.60–1.34 | 0.598 | Random | 66.01 | 0.032 |
| | **Codominant** | AA | 1 | - | - | - | - | - |
| | | AG | - | - | - | - | 75.47 | 0.007 |
| | | GG | - | - | - | - | 83.02 | 0.01 |
| | **Dominant** | AG + GG vs AA | 1.03 | 0.55–1.96 | 0.917 | Random | 67.84 | 0.025 |
| | **Recessive** | GG vs AA + AG | - | - | - | - | 81,9 | 0,009 |
| *FASL* (rs5030772) | **Allelic** | G/A | 1.57 | 0.91–2.71 | 0.108 | Random | 62.96 | 0.061 |
| | **Codominant** | AA | 1 | - | - | - | - | - |
| | | AG | - | - | - | - | 93.61 | 0 |
| | | GG | - | - | - | - | 82.85 | 0.004 |
| | **Dominant** | AG +GG vs AA | - | - | - | - | 94.1 | 0 |
| | **Recessive** | GG vs AA + AG | 0.524 | 0.16–1.70 | 0.282 | Random | 55.72 | 0.107 |
| *CTLA4* (rs231775) | **Allelic** | G/A | 1.1 | 0.98–1.24 | 0.108 | Fixed | 36.67 | 0.243 |
| | **Codominant** | AA | 1 | - | - | - | - | - |
| | | AG | 0.72 | 0.39–1.35 | 0.304 | Random | 64.7 | 0.056 |
| | | GG | 0.89 | 0.54–1.49 | 0.663 | Fixed | 0 | 0.935 |
| | **Dominant** | AG + GG vs AA | 0.76 | 0.46–1.25 | 0.276 | Random | 49.9 | 0.136 |
| | **Recessive** | GG vs AA + AG | 1.08 | 0.68–1.69 | 0.751 | Fixed | 4.85 | 0.338 |

**p- value het**: chi square p value for heterogeneity.

Meta-analysis was not carried out when heterogeneity was greater than 75%.

*Statistically significant result.

The combined analysis for the homozygous codominant (*GG* vs *AA*), heterozygous codominant (*AG* vs *AA*), dominant (*AG* + *GG* vs *AA*) and recessive (*GG* vs *AA* + *AG*) models did not show any significant association (Table 4).

### Association analysis of the IL2RA gene with the risk of developing alopecia areata

To analyze the association of *IL2RA* rs3118470 polymorphism with the risk of developing AA, the combined analysis for the different genetic models: allelic (*T* vs *C*), homozygous codominant (*CC* vs *TT*), heterozygous codominant (*CT* vs *TT*), dominant (*CT* + *CC* vs *TT*) and recessive (*CC* vs *TT* + *CT* models, was proposed, but it was decided not to perform the meta-analysis due to the significant heterogeneity detected between genetic models.

## Discussion

### Genetic and pathophysiological bases of alopecia areata

AA is a multifactorial disease involving important immunological and genetic components in its pathogenesis, with a critical role played by both CD4+ and CD8+ T lymphocytes [1, 3]. Data

from experimental studies have postulated CD8[+] and NKG2D[+] T lymphocytes as fundamental elements for the collapse of immune privilege of the hair follicle through the production of interferon-gamma, triggering an increase of IL-15 and a type I autoimmune reaction [26].

Given the polygenic nature of AA, genome-wide association studies (GWAS) have provided evidence on the involvement of genes related to both innate and adaptive immunity [5]. Variants associated with the development of AA were identified in at least 139 genes. The most relevant genes are associated with antigen presentation (*HLA-DRA*, *HLA-DQA1*, *HLA-DQA2*, *HLA-DQB2* and *HLA-A)*, with intracellular T lymphocyte signaling (*PTPN22*), encoding interleukins related to proliferation of T lymphocytes (*IL-21* and *IL-2*), interleukin receptors (*IL2RA*), inducers of T cell differentiation (*NOTCH-4*), costimulatory molecules (*CTLA4* and *ICOS*), the autoimmune response regulator gene (*AIR*), among other genes such as apoptosis and autophagy regulators (*ACOXL/BCL2L11*), as well as (*FAS/FASL*) [5, 10, 21, 27–29].

## Association of *PTPN22* with alopecia areata

The *PTPN22* gene encodes lymphoid tyrosine phosphatase (LYP) protein, which has been associated with several autoimmune diseases. This protein plays a role in the downregulation of the T cell receptor (TCR) and is essential in proliferation and maturation processes [30, 31]. LYP protein is potentiated by the C-terminal Src kinase (CSK) protein to generate the dephosphorylation of the T cell-specific tyrosine kinase (LCK) protein, and zeta chain of receptor associated kinase 70 (ZAP-70) protein, disrupting the TCR signaling cascade [30, 32].

Genetic variant rs2476601 polymorphism (involves the substitution of arginine for tryptophan, altering the P1 portion of LYP protein, whose function directly affects the interaction with the CSK protein altering the negative regulatory process made by the TCR, as there are fewer LYP-CSK complexes [33, 34].

In the present study, the meta-analysis data generated statistically significant results for the rs2476601 variant of the *PTPN22* gene and its association with the risk of developing AA in allelic models (OR1.49 [95% CI: 1.3–1.95]), heterozygous codominant (OR 1.44 [95% CI:1.19–1.76]) and dominant (OR1.43 [95% CI,1.18–1.73]), indicating that the presence of at least one T allele confers susceptibility to AA. These results are consistent with what was established by Kemp et al. [17] who conducted the first case-control study in a European population, finding the association of the *T* allele with severe forms of AA (OR1.89 [95% CI:1.17–3.05) in a group of 196 cases and 507 controls of English origin. Likewise, a systematic review by Lei et al. [35] determined the protective effect of the *C* allele (OR 0.77 [95% CI: 0.64–0.92]) in an allelic model using the data extracted from 5 primary studies. Similarly, they showed, by comparing the *CC* vs *CT* + *TT* genotypes, that carrying the CC genotype was associated with a decreased risk in the development of AA (OR 0.93 [95% C: 0.60–0.88]) [35].

The relationship with the variant (rs2476601) has been studied in various autoimmune diseases such as type-1 diabetes mellitus, finding that the *T* allele has been identified as a risk factor in North American and Italian populations [33]. In rheumatoid arthritis patients with positive rheumatoid factor, an OR of 1.5 [95% CI:1.1–1.9]), corresponding to the effect of the *T* allele, was found in an English population [36]. A systematic review carried out by Lea and Lee in systemic lupus erythematosus (SLE) patients had an OR of1.56 [95% CI: 1.33–1.82]) in the meta-analysis for the *T* allele, with primary studies including mainly European and Hispanic populations [37]. These findings showed that the *PTPN22* gene plays an important role in the regulation of immune homeostasis. Therefore, it is important to strengthen the knowledge on the frequency of the functional genetic variant rs2476601 and its role in the susceptibility to develop AA, considering aspects related to ethnicity, environmental factors, and

geographic region of the population, given the heterogeneity found in studies reported in the literature. Large-sample studies are strongly recommended.

## Association of *CTLA4* with alopecia areata

The CTLA4 receptor is present in both CD4$^+$ and CD8$^+$ T cells. This protein is important for regulating immunity and maintaining immune tolerance [24, 38]. CTLA4 negatively regulates T lymphocytes by binding to protein B7, which is expressed by the antigen-presenting cell. Activation of the T lymphocyte involves the interaction of the TCR (T cell receptor) with the major histocompatibility complex loaded with the antigen on the surface of APC cells (antigen-presenting cells), but additionally requires co-stimulating signals that enhance the immune response or co-repressing signals that decrease this response (immune checkpoint). The co-stimulatory signal is given by the binding of the B7 protein (APC) to its ligand CD28 in the lymphocyte receptor, which increases the production of IL-2, proliferation, and the survival of the T lymphocyte [38, 39]. Co-repressing signals must be activated to maintain immunological homeostasis. CTLA 4, which is a co-inhibitory protein, competes with CD28 for binding to B7, when the CTLA4/B7 interaction is established, a signal that downregulates the T lymphocyte response is generated [11, 38, 39].

Alterations in the gene that encodes the CTLA4 receptor can trigger lymphocytic autoreactivity that has been postulated in AA pathogenesis, and CTLA4-mediated signaling plays an important role in preventing hair follicle immune privilege collapse [5, 24, 40]. The functional variant rs231775 (+49*G/A*) in the *CTLA4* gene produces a change from alanine to threonine in position 49 of exon 1 [41], which in turn generates an increase of the expression of CTLA4 in the cytoplasmic membrane, altering immunological homeostasis [42]. This variant, like the *PTPN22* gene, has been associated with susceptibility to the development of autoimmune diseases such as systemic lupus erythematosus, type-1 diabetes mellitus, and rheumatoid arthritis. A systematic review performed by Wang et al. suggests the use of rs231775 as a marker of susceptibility for the development of autoimmune diseases in Asian and Caucasian populations [43]. *CTLA4* upregulation has also been evidenced in neoplastic diseases such as pancreatic cancer [44].

The present study did not find a significant association between rs231775 polymorphism of the *CTLA4* gene with the development of AA. However, the studies showed heterogeneous results. John et al. [24] studied a great variety of genetic polymorphisms in *CTLA4* in Central Europe, but in the case of rs231775, the G allele was correlated with the development of AA (OR 1.26 [95% CI:1.12–1.41]), higher disease severity (OR1.43 [95% CI:1.24–1.64]) and early onset of the disease (OR1.39 [95% CI:1.20–1.61]) [24]. Megiorni et al. found no association between the rs231775 variant and the risk of developing AA, in an Italian population [12]. Ismail et al. found a protective effect on the *G* allele (OR: 0.44, 95% CI: 0.23–0.85) when compared to the homozygous form of the *A* allele (*GG + AG* vs *AA*), in an Egyptian male population [23] Finally, Salinas-Santander et al. study in a Mexican population, found no association between rs231775 and rs3087243 polymorphisms and the development of AA [25]. The later demonstrates that the rs231775 and rs3087243 variants of the CTLA4 gene play an important role in the pathophysiology of AA, however, they must be carefully interpreted according to the origin of the population.

## Association of *IL2RA* with alopecia areata

The *IL2RA* gene encodes the alpha chain of the IL-2 receptor (also known as CD25), which makes up one of the three receptor subunits and confers its high affinity to IL-2 on effector and regulatory T cells (CD4$^+$ CD25$^+$ FOXP3$^+$) [45, 46]. These cells suppress autoreactive T

lymphocytes and require IL-2 for their proper development and homeostasis, which is why alterations in this gene are correlated with immunodeficiency [47] or the development of auto-immune diseases such as type-1 diabetes mellitus, multiple sclerosis, systemic lupus erythematosus, rheumatoid arthritis and celiac disease [5, 48–50].

The rs3118470 (*T>C*) polymorphism of *IL2RA*, corresponds to an intronic variant located at the 5' end of intron 1, the biological mechanism of the association of this variant with various diseases has not yet been determined. Separated at 3kb from this variant, another linkage disequilibrium polymorphism (rs706778) has been identified and has been associated with the development of type 1 diabetes mellitus [51].

In the present study, it was not possible to compute the combined measure of the effect of the included studies, given the substantial inter-study heterogeneity identified, which was mainly influenced by the clinical heterogeneity attributed to the fact that the studies were carried out in populations located in different geographical regions and a limited population in the included studies. The analysis by types of alopecia could not be performed since not all studies reported this feature.

Regarding individual results, Redler et al. study, in German and Belgium population, identified the *C* allele of the *IL2RA* rs706778 gene as a risk factor for developing AA (OR 1.3 [5% CI:1.12–1.51]), with a more severe form of the disease (OR 1.45 [95% CI,1.22–1.73]) in patients with a family history of AA (OR1.4 [95% CI:1.11–1.78]) [22]. Miao et al. study, in a Chinese population, found significant differences between frequencies of the *C* and *T* alleles; between cases and controls ($p < 0.0001$). The study also evidenced that the allele and genotypic frequencies among the groups of severe and mild alopecia areata *p*-values results were not significant, as determined by the authors of the studies included in this systematic review ($p = 0.289$ and $p = 0.137$, respectively) [13]. Moravvej *et al.* considered the presence of the *C* allele (OR 3.56 [95% CI,1.89–6.71]) as a risk factor for developing AA [4].

## Association of *FAS/FASL* with alopecia areata

FAS/FASL pathway is critical for maintaining immunological homeostasis [52], due to the ability to induce programmed cell death in T cells, its role in the proliferation, activation and differentiation of T cells such as Th17 [6, 52–54]. In the AA context, it has been determined that variations in *FAS/FASL* genes could affect the apoptosis of T lymphocytes and natural killer cells involved in the pathogenesis of the disease [55]. Experimental studies have determined that FAS is expressed in hair follicles and FASL in perifollicular inflammatory infiltrate cells [56]. Therefore, as the FAS/FASL pathway is involved in the co-stimulation of both CD4[+] and CD8[+] during the early phases of immune response inducing apoptosis in follicular keratinocytes, it explains the absence of inflammatory infiltrates and resistance to AA development in mouse models with FAS/FASL deficiency [56, 57].

Functional rs1800682 polymorphism (−670 *A>G*) is found in the *FAS* gene promoter [58], specifically in the STAT1 (signal transducer and activator of transcription 1) binding site, a key element to initiate the process of transcriptional activation and expression of the *FAS* gene [59, 60]. Similarly, the rs5030772 variant (IVS2nt-124 A> G) is located in intron 2 and plays an important role in the transcription and expression of the *FASL* gene [58, 61].

The variants in the *FAS* and *FASL* genes did not show significant results in the present study. A meta-analysis was not performed for different genetic models since a high inter-study heterogeneity was identified. This could be explained by the small number of studies and sample size. However, rs1800682 has been reported to be associated with susceptibility to the development of AA. Kalkan *et al.* [10] identified allele *A* as a risk factor (OR 1.20 [95% CI: 0.82–1.75]) and *GG* genotype as a protective factor when comparing with *AG* and *AA*

genotypes (OR 0.07 [95% CI: 0.00–0.4]) of the *FAS* rs1800682 variant. No significant findings were found for the rs5030772 variant of *FASL* gene [10]. In the study conducted by Fan *et al.* [19], a protective effect for the *GA* genotype when compared with the AA genotype (OR 0.43 [95% CI: 0.22–0.86]) was identified, and the homozygous form of this allele (*GG* genotype) showed a protective effect (OR 0.61 [95% CI, 0.23–0.86]). The effect of the 844*T>C* variant on the *FASL* gene was also assessed and no association with a combination with variants in the *FAS* gene was found [19].

Tabatabaei-Panah *et al.* [21] found no effect of *FAS* variant rs1800682 on the risk of developing AA, however, they did find an effect of both allele *A* (OR 2.36 [95% CI:1.21–4.59]) and genotype *AA* (OR 2.13 [95% CI:1.12–4.03]) of the rs5030772 variant of the *FASL* gene [21]. Finally, in a study carried out by Seleit *et al.* [20] in an Egyptian population, the presence of a *G* allele of the *FAS* variant rs1800682 conferred a risk effect (OR 1.75 [95% CI: 1.42–2.33]) as well as the homozygous form (*GG* genotype) of the same variant (OR 5.1 [95% CI: 1.25–20.48]), contrary to what was found by Kalkan *et al.* [10], who found a protective effect of the *GG* genotype in the Turkish population (OR 0.07 [95% CI: 0.00–0.4]) compared to *AG* + *AA* genotypes [10]. This heterogeneity may be influenced by geographical factors.

## Certainty of the evidence according to the GRADE approach

The results obtained by using the GRADEpro tool, made it possible to analyze the evidence for the allelic, codominant heterozygous and dominant genetic models, graded as moderate and high quality respectively regarding evidence on the *PTPN22* gene. This outcome was consistent with the individual evaluation of the studies using the NOS, in which individual studies obtained high scores. Although observational studies are known to have low quality of evidence, an adequate design and execution of primary studies allow an increase in the degree of certainty of the evidence.

## Limitations of the present study

The outcomes obtained in this study evidenced several limitations which must be considered for data interpretation. Regarding the *PTPN22* gene, it was not possible to analyze the subgroups by ethnicity, which usually allows a better interpretation of the data of multifactorial diseases such as AA. Furthermore, it was only possible to perform a meta-analysis for one variant per gene from the data obtained in the extraction phase, since not all studies analyzed the association between AA and the same gene polymorphisms. For that reason, we could not identify the contribution of each variant to the susceptibility to develop the disease.

Haplotype analysis may have provided more information regarding the role played by the variants in AA risk. On the other hand, the number of studies included for the combined analysis of each gene was limited, which reduces the statistical power of the estimates and allows us to identify that the estimates could be altered due to publication bias. Some of the analyses proposed could not be carried out since all the included studies did not consider the same variables, such as the different forms of AA.

The results obtained from the available studies for the genes included in this systematic review, evidence the need of further genetic association studies regarding genes that encode proteins that participate in the immunological pathway related to AA, in order to clarify the role that these variants have in the susceptibility to develop AA.

## Conclusion

This study confirms the association of the *PTPN22* gene rs2476601 variant with the risk of developing AA. This is evident in the increased risk that patients carrying the *T* allele have

when compared to carrying the *C* allele. However, to perform more robust studies identifying the ethnic background of the population of origin, is required, so that the risk identified in the present study can be validated.

A statistical study in greater depth, on the effect of variants in the *CTLA4*, *FAS*/*FASL* and *IL2RA* genes, requires conducting a greater number of genetic association studies in order to calculate a combined effect measure.

## Supporting information

**S1 File. PRISMA 2009 checklist.**
(DOC)

**S2 File. Research strategy.**
(DOCX)

**S3 File. Quality of evidence assessment using GRADEpro tool.**
(DOCX)

**S4 File. Forest plot performed for the PTPN22 gene.** PTPN22.
(DOCX)

**S5 File. Forest plot performed for the *FAS* gene.**
(DOCX)

**S6 File. Forest plot performed for the *FASL* gene.**
(DOCX)

**S7 File. Forest plot performed for *CTLA4* gene.**
(DOCX)

## Author Contributions

**Conceptualization:** S. R. Gil-Quiñones, L. D. Gutierrez-Castañeda.

**Data curation:** S. R. Gil-Quiñones, I. T. Sepúlveda-Pachón, L. D. Gutierrez-Castañeda.

**Formal analysis:** S. R. Gil-Quiñones, I. T. Sepúlveda-Pachón, G. Sánchez Vanegas.

**Investigation:** S. R. Gil-Quiñones, I. T. Sepúlveda-Pachón, G. Sánchez Vanegas, L. D. Gutierrez-Castañeda.

**Methodology:** G. Sánchez Vanegas, L. D. Gutierrez-Castañeda.

**Software:** G. Sánchez Vanegas.

**Supervision:** G. Sánchez Vanegas, L. D. Gutierrez-Castañeda.

**Validation:** L. D. Gutierrez-Castañeda.

**Writing – original draft:** S. R. Gil-Quiñones, I. T. Sepúlveda-Pachón, L. D. Gutierrez-Castañeda.

**Writing – review & editing:** S. R. Gil-Quiñones, I. T. Sepúlveda-Pachón, G. Sánchez Vanegas, L. D. Gutierrez-Castañeda.

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
