## [Decision Letter · Decision Letter 0]

29 Jun 2021

PONE-D-21-16743

Effect of polymorphisms on PTPN22, FAS/FASL, IL2RA and CTLA4 genes on the risk of developing alopecia areata. Systematic review of literature and meta-analysis

PLOS ONE

Dear Dr. Gutierrez-Castañeda,

Thank you for submitting your manuscript to PLOS ONE. After careful consideration, we feel that it has merit but does not fully meet PLOS ONE’s publication criteria as it currently stands. Therefore, we invite you to submit a revised version of the manuscript that addresses the points raised during the review process.

After a carefull consideration of the Reviewers' comments, I suggest to revise the manuscript according to Reviewer1 suggestions. Moreover, it is mandatory to pay close attention to English and grammar.

We look forward to receiving your revised manuscript.

Kind regards,

Cinzia Ciccacci

Academic Editor

PLOS ONE

Journal Requirements:

3. Please remove your figures from within your manuscript file, leaving only the individual TIFF/EPS image files, uploaded separately.  These will be automatically included in the reviewers’ PDF.

5. We note that this manuscript is a systematic review or meta-analysis; our author guidelines therefore require that you use PRISMA guidance to help improve reporting quality of this type of study. Please upload copies of the completed PRISMA checklist as Supporting Information with a file name “PRISMA checklist”.

Reviewers' comments:

Reviewer's Responses to Questions

**Comments to the Author**

1. Is the manuscript technically sound, and do the data support the conclusions?

Reviewer #1: Yes

Reviewer #2: Yes

2. Has the statistical analysis been performed appropriately and rigorously? 

Reviewer #1: Yes

Reviewer #2: Yes

3. Have the authors made all data underlying the findings in their manuscript fully available?

Reviewer #1: Yes

Reviewer #2: Yes

4. Is the manuscript presented in an intelligible fashion and written in standard English?

Reviewer #1: No

Reviewer #2: No

5. Review Comments to the Author

Reviewer #1: Authors have investigated the polymorphisms of the candidate genes in susceptibility to alopecia areata in a meta-analysis study. They have indicated that rs2476601 variation in the PTPN22 gene is a risk factor for the development of alopecia areata. The authors did a nice job to summarize the previous studies regarding the influence of genetic alteration in susceptibility to alopecia areata. I have only some minor comments:

- Although the discussion part is categorized to explain all the variations in several populations with different ethnicity, this part needs to be improved by making a short conclusion at the end of each section. Only explaining the results of the previous studies is not enough for a systematic review of the literature. In the present form, the authors mostly stated the results which some of which are significant in the specific populations. However, at the end of the study, they conclude that only a variation in the PTPN22 gene is associated with the disease.

- The authors stated in the title of the study "Effect of polymorphisms on PTPN22, FAS/FASL, IL2RA and CTLA4 genes on the risk of developing alopecia areata". Are the authors are evaluating the influence of the polymorphisms on the genes (it seems that there is no data available reporting gene alteration) or the disease development. If the second is correct (which it seems that this is the case), the title needs to be revised.

- The English language needs to seriously be improved.

Reviewer #2: the manuscript generally good, I did not find dual publication. I detected that the research ethics are provided in the manuscript. the manuscript need English and good grammar editing, it should revised by native English speaker.

6. PLOS authors have the option to publish the peer review history of their article (what does this mean?). If published, this will include your full peer review and any attached files.

Reviewer #1: No

Reviewer #2: No

---

## [Author Response · Author response to Decision Letter 0]

26 Aug 2021

Dear editor:

Thank for the review. 

We send the required answers

- There are many language and grammar errors and need to be edited correctly. 

Answer: the manuscript has been fully corrected 

- There are minor revisions as follow: 

- Line 65, the word its should be their. 

Answer: the correction has been made 

- Line 66, function of what? 

Answer: the sentence has been changed and explained 

- Line 77, the word report should be reported

Answer: the word has been corrected

- Line 84, (Or=0 .28, 95% CI 0.09-0.82) is the same for allele A (rs 231775) and G (rs 3087243)?. 

Answer: In the paragraph, the OR for the haplotype containing these two alleleshas been estimated. The sentence was corrected

- Line 92, its should be their.

Answer: error has been corrected 

- Line 109, Gen is the abbreviation of what?: 

Answer: the correct word is Gene; the word has been corrected 

- Line 170, rs of FAS gene is written incorrectly. 

Answer: rs for the FAS gene has been corrected 

- Line 171, the authors should write rs of CTLA4 gene.

Answer: CTLA4 gene rs has been corrected 

- In statistical analysis part and other parts, the word analyzes is wrong. This word is verb and not noun.

Answer: the word analyzes has been corrected, it has been changed by analysis. 

- In table 3, why the authors write allele ½ in the head of study information column and write allele 2 in head of cases column.

Answer: allele number 1 alluded to the major allele and allele number 2 to the minor allele. The correct name has been corrected in the table. 

- In line 221, Rs should be rs

Answer: the correction has been made 

- Replace each word alele by word allele.

Answer: the correction has been made

 - In line 280, LYP ptn, the abbreviation of what? 

Answer: the abbreviation has been described in the paragraph

- Demonstrate the abbreviated LCK protein and CSK terms.

Answer: the abbreviations has been described in the paragraph

- The paragraph included in lines 286 to 290 in discussion need more clarification.

Answer: The information in the paragraph has been expanded.

- Line 366, C>T or T>C. it should T>C. 

Answer: the correct manner is T>C. It has been corrected.

- Line 384 in discussion part, p values results not significant as authors determined. 

Answer: the information in the paragraph has been corrected

- Line 434, the word a priori word is wrong.

Answer: the paragraph was restructured

Reviewers' comments:

Reviewer's Responses to Questions

Comments to the Author

1. Is the manuscript technically sound, and do the data support the conclusions?

Reviewer #1: Yes

Reviewer #2: Yes

2. Has the statistical analysis been performed appropriately and rigorously?

Reviewer #1: Yes

Reviewer #2: Yes

3. Have the authors made all data underlying the findings in their manuscript fully available?

Reviewer #1: Yes

Reviewer #2: Yes

4. Is the manuscript presented in an intelligible fashion and written in standard English?

Reviewer #1: No

Reviewer #2: No

Answer: The manuscript was revised again in order to correct English grammar errors.

5. Review Comments to the Author

Reviewer #1: Authors have investigated the polymorphisms of the candidate genes in susceptibility to alopecia areata in a meta-analysis study. They have indicated that rs2476601 variation in the PTPN22 gene is a risk factor for the development of alopecia areata. The authors did a nice job to summarize the previous studies regarding the influence of genetic alteration in susceptibility to alopecia areata. I have only some minor comments:

- Although the discussion part is categorized to explain all the variations in several populations with different ethnicity, this part needs to be improved by making a short conclusion at the end of each section. 

Answer: the explanation was supplemented at the end of each paragraph

Only explaining the results of the previous studies is not enough for a systematic review of the literature. In the present form, the authors mostly stated the results which some of which are significant in the specific populations. However, at the end of the study, they conclude that only a variation in the PTPN22 gene is associated with the disease.

Answer: According to the analysis carried out, it was found that only the PTPN22 gene has statistical significance. Regarding to the other genes, some studies show statistical significance, however when all the studies are analyzed together (as a meta analysis) and the quality of the studies is evaluated, their role in the risk of developing alopecia cannot be demonstrated.

- The authors stated in the title of the study "Effect of polymorphisms on PTPN22, FAS/FASL, IL2RA and CTLA4 genes on the risk of developing alopecia areata". Are the authors are evaluating the influence of the polymorphisms on the genes (it seems that there is no data available reporting gene alteration) or the disease development. If the second is correct (which it seems that this is the case), the title needs to be revised.

Answer: The title was corrected. We evaluated the influence of the polymorphisms on the alopecia areata development.

- The English language needs to seriously be improved.

Answer: The manuscript was revised again in order to correct English grammar errors.

Reviewer #2: the manuscript generally good, I did not find dual publication. I detected that the research ethics are provided in the manuscript. the manuscript need English and good grammar editing, it should revised by native English speaker.

Answer: The manuscript was revised again in order to correct English grammar errors.

---

## [Editor Report · Decision Letter 1]

29 Sep 2021

Effect of PTPN22, FAS/FASL, IL2RA and CTLA4 genetic polymorphisms on the risk of developing alopecia areata: A systematic review of the literature and meta-analysis.

PONE-D-21-16743R1

Dear Dr. Gutierrez-Castañeda,

We’re pleased to inform you that your manuscript has been judged scientifically suitable for publication and will be formally accepted for publication once it meets all outstanding technical requirements.

Kind regards,

Cinzia Ciccacci

Academic Editor

PLOS ONE
---

## [Editor Report · Acceptance letter]

5 Oct 2021

PONE-D-21-16743R1 

Effect of PTPN22, FAS/FASL, IL2RA and CTLA4 genetic polymorphisms on the risk of developing alopecia areata: A systematic review of the literature and meta-analysis 

Dear Dr. LD:

I'm pleased to inform you that your manuscript has been deemed suitable for publication in PLOS ONE. Congratulations! Your manuscript is now with our production department. 

Kind regards, 

on behalf of

Dr. Cinzia Ciccacci 

Academic Editor

PLOS ONE